# Vitamin D and Immunological Patterns of Allergic Diseases in Children

**DOI:** 10.3390/nu13010177

**Published:** 2021-01-08

**Authors:** Agnieszka Lipińska-Opałka, Agata Tomaszewska, Jacek Z. Kubiak, Bolesław Kalicki

**Affiliations:** 1Pediatric, Nephrology and Allergology Clinic, Military Institute of Medicine, 04-141 Warsaw, Poland; awawrzyniak@wim.mil.pl (A.T.); kalicki@wim.mil.pl (B.K.); 2Laboratory of Regenerative Medicine and Cell Biology, Military Institute of Hygiene and Epidemiology (WIHE), 04-141 Warsaw, Poland; jacek.kubiak@univ-rennes1.fr; 3Cell Cycle Group, Faculty of Medicine, Institute of Genetics and Development of Rennes, University of Rennes 1, UMR 6290, CNRS, Rennes, France

**Keywords:** vitamin D, asthma, atopic dermatitis, regulatory T cells, children

## Abstract

Vitamin D, in addition to its superior role as a factor regulating calcium-phosphate metabolism, shows wide effects in other processes in the human body, including key functions of the immune system. This is due to the presence of vitamin D receptors in most cells of the human body. In our study, we aimed to assess whether there is a correlation between vitamin D content and the clinical course of allergic diseases as well as establish their immunological parameters in children. We found that vitamin D deficiency was significantly more frequent in the group of children with an allergic disease than in the control group (*p* = 0.007). Statistically significant higher vitamin D concentrations in blood were observed in the group of children with a mild course of the disease compared to children with a severe clinical course (*p* = 0.03). In the group of children with vitamin D deficiency, statistically significant lower percentages of NKT lymphocytes and T-regulatory lymphocytes were detected compared to the group of children without deficiency (respectively, *p* = 0.02 and *p* = 0.05), which highlights a potential weakness of the immune system in these patients. Furthermore, statistically higher levels of interleukin-22 were observed in the group of children with vitamin D deficiency (*p* = 0.01), suggesting a proinflammatory alert state. In conclusion, these results confirm the positive relationship between the optimal content of vitamin D and the lesser severity of allergic diseases in children, establishing weak points in the immune system caused by vitamin D deficiency in children.

## 1. Introduction

Vitamin D is a long known factor regulating calcium phosphate metabolism in the human body. It has gained a broader meaning in the last two decades, mostly due to the discovery of its pleiotropic effects on the human body, resulting from the presence of vitamin D receptors (VDR) on cells belonging to many different tissues and systems. The presence of VDR was found among others in the brain, eyeballs, heart, β cells of pancreatic islands, muscles, adipose tissue, parathyroid glands, adrenal glands and, most importantly for our research, in almost all cells of the immune system [1]. The role of vitamin deficiency D in the development of many diseases, including allergic diseases, as well as the potential benefits of its supplementation in their treatments, is currently the subject of numerous scientific studies, and there remains no consensus in this field.

Vitamin D occurs in two major forms: D2 (ergocalciferol) and D3 (cholecalciferol) in living organisms. Vitamin D2 is formed after UVB exposure of ergosterol produced by mushrooms and plankton, whereas D3 is synthesized endogenously as a product of the photochemical transformation of 7-dehydrocholesterol in the human skin, also found naturally in oily fish [2]. These two forms of vitamin D undergo the same metabolic processes in the human body.

Previtamin D3 (pre-D3) is formed in the skin under UVB radiation (290–315 nm wavelength) from 7-dehydrocholesterol (7-DHC) [3]. During this process, the B-ring of 7-DHC is photolyzed. Prolonged UVB exposure leads to the photochemical transformation of pre-D3 to lumisterol (L3), which involves the resealing of the B-ring, albeit differently than the initial configuration, making L3 a stereoisomer of 7-DHC. It was believed for many years that this is a kind of self-regulating process occurring in such a way that the excessive sunlight inactivates pre-D3 and thus protects against intoxication with vitamin D3. However, it has recently been demonstrated that L3 can be enzymatically hydroxylated to produce several biologically active products detectable in the human body that can interact with the nongenomic binding site of the VDR molecule [4].

Vitamin D (both as vitamin D2 and D3), after entering circulation, is firstly hydroxylated in the liver by vitamin D-25-hydroxylase (CYP2R1) to 25-hydroxyvitamin D [25(OH)D, calcidiol] and next, mostly in kidneys, by D-1α-hydroxylase (CYP27B1) to the active form of vitamin D-1, 25-dihydroxyvitaminum D [1,25(OH)2D]. However, apart from kidneys, CYP27B1 is expressed in many other tissues. Thus, the active form of vitamin D can be formed in other organs of the human body. Calcitriol acts on target tissues through specific VDR receptors, which function as transcription factors [2].

In addition to this classic pathway, it was recently demonstrated that both vitamin D2 and vitamin D3 can be activated through an alternative pathway by the CYP11A1 enzyme, mostly in the skin, placenta, adrenal glands and immune cells [5]. The main products of the above vitamin D metabolism can also be hydroxylated in position 1α by CYP27B1, albeit with significantly lower efficiency than 25(OH)D3. Final products of the CYP11A1-conducted reaction are biologically potent to a similar extent to the classical 1,25(OH)2D3-proceeded reaction to antiproliferative, prodifferentiation, antifibrotic and antineoplastic properties, while being noncalcemic at doses far above therapeutic limits for 25(OH)D3 and 1,25(OH)2D3 [6,7]. Furthermore, these metabolites, as well as 1,25(OH)2D3, act not only through VDR but also as antagonists or inverse agonists of the retinoic acid-related orphan receptors α–γ (RORα–γ; NR1F1–3), members of the ROR subfamily of nuclear receptors. Throughout these pathways, they play a critical role in the regulation of a number of physiological processes, including several key immune functions [8]. Moreover, one of the metabolites of the enzyme CYP11A1, 1,23(OH)2D3, has recently been shown to act mainly through the arylhydrocarbone receptor (AhR), which is another alternative pathway for vitamin D3-hydroxyderivatives [9].

Skin synthesis is a well-known and important source of vitamin D. However, its efficiency can be reduced by up to 95–98% after the topical application of a sunscreen with an SPF of 30 [3]. In Central and Eastern Europe, adequate endogenous synthesis occurs from April to September with sun exposure for a minimum of 15 min on 18% body surface (exposed forearms and partly legs) without using creams with a filter. The second source of vitamin D is oral intake. It can be supplied with animal (fish oil, fatty fish, liver, eggs) or vegetable (dried mushrooms) products [2].

The degree of vitamin D body supply is determined based on the serum 25-hydroxyvitamin D concentration. This hepatic metabolite of vitamin D has a longer half-life (about three weeks) compared to 1.25 (OH) D (4–6 h). It reaches over 1000-fold higher serum concentrations, and its synthesis is not regulated by calcium and phosphate metabolism [10]. Table 1 presents the accepted standards for vitamin D concentrations [11].

Great emphasis is currently placed on maintaining normal vitamin D levels in both pregnant women and children, among others, due to the role of vitamin D in the development and course of allergic diseases. The problem of allergic diseases has grown in recent years. Large cities in Poland have expanded, resulting in increased allergic disease incidence. The rate of atopic dermatitis (AD) incidence among children in Poland ranges from 4.7 to 9.2% [12], and even 16–19% of children suffer from asthma [13]. At the root of both these diseases are, among others, disorders of the immune system, and the latter, in turn, is influenced by vitamin D. Vitamin D was shown to inhibit Th1 cells by reducing the production of proinflammatory cytokines, such as IL-2, INF-gamma, TNF-alpha and others [14]. On the other hand, the effect of vitamin D on the Th2 lymphocyte system is still unclear. However, some in vitro studies have shown that it can induce a Th2-mediated inflammatory response by increasing the expression of cytokines IL-4, IL-5 and IL-13 [15]. For these reasons, we undertook our study to estimate the influences of vitamin D levels on allergic diseases in pediatric patients in Warsaw, Poland and clarify the potential role of the cytokines involved in allergic diseases.

## 2. Materials and Methods

### 2.1. Analyzed Patients

Seventy-five children with atopic dermatitis or/and asthma treated at the Department of Pediatrics, Nephrology and Allergology of the Military Institute of Medicine in Warsaw during the period of 2015–2017 qualified for this study. The median age in this group was 7 years (IQR: 3–10). The diagnosis of AD was based on the criteria of Hanifin and Rajka [16], and the diagnosis of asthma was based on Global Initiative for Asthma (GINA) 2014 [10]. In the next stage, some parameters of the atopic background of the disease (total serum IgE and skin prick tests) were confirmed in order to narrow down the study group. 

The control group represented 37 children without symptoms of allergic disease and without signs of respiratory tract infections (median age: 8; IQR: 6–11). Each time parents of patients signed the written consent to the study. The research project was approved by the Bioethics Committee at the Military Medical Chamber (Resolution No. 123/14 of 11 April 2014).

### 2.2. Course of the Disease

Asthma severity was assessed retrospectively from the level of treatment required to control symptoms and exacerbations. Patients with asthma were divided into three groups:

Asthma severity was assessed retrospectively from the level of treatment required to control symptoms and exacerbations, which is the method recommended by GINA experts. 

For the purposes of the study, patients with asthma were divided into three groups:Mild asthma—asthma controlled with Step 1 or Step 2 treatment;Moderate asthma—asthma controlled with Step 3 treatment;Severe asthma—asthma controlled with Step 4 or Step 5 treatment [17].

Step 1 treatment included only short-acting beta-agonists (SABA) “on-demand” treatment, without inhaled glucocorticosteroids (GCS). In Grade 2, treatment was based on small doses of the inhaled GCS and SABA “on demand.” In patients with Grade 3, low doses of inhaled GCS were used with long-acting beta-agonists (LABA). At Stage 4, a combination of medium-inhaled doses of GKS/LABA with SABA “on demand” or high doses of inhaled GCS were used. In Grade 5, anti-IgE therapy was considered.

The severity of atopic dermatitis was determined using the Scoring Atopic Dermatitis Scale (SCORAD). This scale determines the severity of AD on the basis of objective symptoms (the extent of the lesions and their severity) and subjective symptoms (pruritus and sleep disturbances) experienced by the patient. The maximum number of points for the patient is 103, and the minimum is 0. A score of 0 means that the subject does not currently have atopic changes or any other symptoms of the disease. The patients with AD were divided into three groups:Mild atopic dermatitis—SCORAD < 20;Moderate atopic dermatitis—SCORAD 20–40;Severe atopic dermatitis—SCORAD > 40.

Finally, for the purposes of this study, both asthma and atopic dermatitis were treated together as “allergic diseases.” All children from the study group were divided into 3 subgroups (combined for asthma and AD): children with mild, moderate and severe courses of allergic disease.

### 2.3. Vitamin D Blood Concentration

Blood was collected for ethylenediaminetetraacetic acid (EDTA) to determine the concentration of vitamin D. The Dia-Sorin LIAISON^®^ device using the chemiluminescence phenomenon (CLIA) was used for this analysis. Vitamin D concentrations were given in absolute terms in ng/mL. According to recommendations [11], we considered the concentrations of vitamin D in the range from 20 ng/mL to 80 ng/mL as optimal values.

### 2.4. Phenotype of Peripheral Blood Lymphocytes

#### 2.4.1. Phenotype of Peripheral Blood Lymphocytes

Samples of 100 µL of blood were collected into ethylenediaminetetraacetic acid (EDTA)-primed tubes. They were incubated for 20 min with 20 µL of primary antibodies against the following lymphocytic antigens: CD3+/19+, CD4+/CD8+, CD19+, CD16/56, NKT, HLA-DR-specific CD3, using an IMK Plus kit (BD Biosciences; Warsaw, Poland). After the erythrocytes’ lysis, cells were washed twice with the phosphate buffer solution and fixed in 1% paraformaldehyde in phosphate buffer. Then, cells were subjected to cytometric analysis by the CellQuest Pro software in a FACS Calibur Flow Cytometer (BD Biosciences; Warsaw, Poland). The results of the lymphocyte phenotypes were presented as mean percentages ± SD.

#### 2.4.2. Natural T-Regulatory Lymphocytes

Twenty microliters of antibodies were added (test sample: CD4-PerCP, CD25-APC, CD127-FITC; isotype control: CD4-PerCP, IgG1 APC, IgG1 FITC) to the cytometric cells. Then, 100 µL of blood was added to each test tube. After 20 min of incubation, the solution for erythrocyte lysis was added, and the whole mixture was incubated for an additional 10 min. Next, it was rinsed with PBS solution and spun down, and the buffer for fixing the cell membrane was added. The total solution was incubated for 40 min. Then, 20 µL of antibodies FoxP3 PE, and to the control sample 20 µL IgG1 PE, was added to the sample. After 40 min of incubation samples were rinsed twice and mixed in 300 µL of 1% solution of paraformaldehyde in PBS. Cytometric analysis using the FACS Calibur Flow Cytometer, USA was performed next. Cells were acquired in flow cytometry. In total, 10,000 counts of CD 4 PerCP-positive cells finished the acquisition. The percentage of nTreg cells (FoxP3+, CD25high, CD127) was counted in CD4-positive cells.

#### 2.4.3. Cytokines

Blood samples were stored at a temperature of −80 °C until the analysis was conducted. After defrosting and thawing at 4 °C, the following ingredients were added to appropriate cytometric tubes: 50 µL of beads, 50 µL of tested serum and 50 µL of detection reagent. The dilutions of tested cytokines were prepared by the same method (standard curve). After that, samples were incubated for 3 h. After that time, each sample was rinsed and spun down. Then, the supernatant was collected, and 300 µL of rinsing buffer was added. The samples were marked with the use of a flow cytometer FACS Calibur. The results were converted to the values from the standard curve and were presented as an average concentration of tested cytokine (pg/mL) ± SD in blood.

### 2.5. Statistical Analysis 

The results were statistically analyzed using the StatSoft software, Inc. software (2014) STATISTICA. The analyzes were initially verified using the normality diagram of the distribution and the Kolmogorov normality test of Smirnov and Liliefors. The Student’s *t*-test was used to evaluate variables with a normal distribution. For the selected variables inconsistent with a normal distribution, nonparametric tests, which do not require normality of distribution (Mann–Whitney U test), were used for statistical evaluation. Correlations were calculated for variables lacking a normal distribution using the Spearman rank factor. Correlations for the variables with a normal distribution were calculated using Pearson’s correlation factor. A value of *p* < 0.05 was considered statistically significant.

## 3. Results

This section may be divided by subheadings. It should provide a concise and precise description of the experimental results, their interpretation as well as the experimental conclusions that can be drawn.

### 3.1. Vitamin D and the Course of the Disease

The mean concentration of vitamin D was statistically significantly lower in the study group compared to the control (27.6 ng/mL ± 11.0 vs. 32.41 ng/mL ± 9.4; *p* = 0.02) (Figure 1). Vitamin D deficiency was significantly more frequent in the study group than in the control (*p* = 0.007) (Figure 2).

After dividing the study group into children with a mild, moderate and severe clinical course, of the allergic disease, we observed that the most numerous group was composed of children with a severe course. A mild clinical course was found in 17 children (23%), moderate in 22 (29%) and severe in 36 cases (48%) (Figure 3).

Statistically significantly higher vitamin D concentrations were observed in the group of children with a mild course of the disease compared to children with a severe clinical course (29.70 ng/mL, q25–20.5, q75–38.9 vs. 21.55 ng/mL, −16.5, q75–30.4; *p* = 0.03) (Figure 4).

### 3.2. Vitamin D and the Immune Profile

The results of the study did not confirm the influence of vitamin D on the concentration of total IgE antibodies and the number of eosinophils (Table 2).

In the group of children with vitamin D deficiency, statistically significantly lower percentages of NKT lymphocytes and T-regulatory lymphocytes were found compared to the group of children with an optimal and suboptimal vitamin D concentration. There were no significant differences in the remaining analyzed parameters. In the control group, there was no correlation between vitamin D levels and the phenotype of peripheral blood lymphocytes (Table 3).

Statistically significantly higher levels of interleukin-22 were observed in the group of children with vitamin D deficiency compared to children with suboptimal and optimal vitamin D concentrations. There was no correlation between the level of vitamin D and the analyzed cytokines in the control group (Table 4).

## 4. Discussion

Vitamin D deficiency is the most common, underdiagnosed and inadequately treated nutritional deficiency in the world [18]. This is mainly due to the very dynamic change in the human lifestyle that has occurred in just a few decades. The current lifestyle is largely associated with spending long working hours inside buildings, which reduces exposure to solar radiation, necessary, in turn, for the natural synthesis of vitamin D in the human skin. Another cause of this deficiency is currently different eating habits, which prevailed half a century ago. The decline in fish consumption and inadequate processing (frying instead of boiling or baking) are the main causes. Finally, for skin cancer prevention, the undoubted benefits of using sunscreen with UV filters also stand in opposition to the natural synthesis of vitamin D in the skin [3].

The results of our study showed that children with allergies had lower average vitamin D levels compared to the group of healthy children. Taking into account the factors mentioned above, it can be suspected that the reason for this may be a lower exposure to solar radiation, but this factor was not assessed in this study. It is the result of more frequent infections and exacerbations of the disease itself, which naturally prevent children from being active in their backyards. An important part of the cause may be provoked by the inhalation of allergens, which makes people prefer to stay at home during the pollen season. Moreover, it should be noted that skin-synthesizing vitamin D is partially damaged in atopic dermatitis, which significantly impairs its function [19]. A meta-analysis of 24 studies conducted by Kim and colleagues in 2016 also concluded that children suffering from atopic dermatitis had lower 25(OH)D levels compared to healthy children (standardized mean difference = −2.03 ng/mL; 95% CI = −2.98 to −1.08). Additionally, the benefits of vitamin D supplementation were assessed in this analysis. Children who received vitamin D had a less severe disease course, assessed by SCORAD or EASI scales [20]. Similarly, in the case of our research, we have shown that children with a more severe allergic disease had lower levels of vitamin D compared to those with a milder course of the disease.

Peroni and coworkers [21] also noticed the correlation between the course of allergic diseases and the serum vitamin D concentration in children suffering from atopic dermatitis. The 25(OH)D concentration was significantly lower in patients with a moderate and severe course of the disease [21]. Sharma [22] described an inverse relationship between the concentration of vitamin D in the blood serum and the SCORAD index in 40 patients between 2 and 18 years of age. Very similar results were obtained by Korean researchers [23] after examining a group of 498 children with atopic dermatitis and Turkish researchers [24] in a group of 60 sick children.

Similarly, in reference to childhood asthma, a group of Turkish researchers noticed a relationship between the concentration of vitamin D and the parameters of the spirometry. They showed a significantly negative correlation between vitamin D levels and the FEV1% change (bronchoreversibility) as well as a significantly positive correlation between max FEV1 reduction and serum vitamin D levels in asthmatic children. They also observed a positive correlation between maximum FEV1 reduction in the exercise test and serum vitamin D levels in the moderate persistent asthma group [25].

Apart from the clinical correlation, many researchers emphasize the role of an adequate supply of vitamin D in the proper functioning of the immune system, and disability is an important component of the pathogenesis of allergic diseases. Vitamin D has been shown to inhibit the secretion of cytokines produced by Th1 and Th17 lymphocytes and stimulate the formation of regulatory T lymphocytes [26]. The results of our study also confirmed these reports. Children from the study group, who had vitamin D deficiency, also had lower levels of regulatory T lymphocytes in the serum compared to the group of children without deficiency. Similarly, Maalmi et al. [27] demonstrated a directly proportional correlation between the percentage of CD25highFoxp3 + Treg cells and vitamin D values in the blood of asthmatic patients (r = 0.368; *p* = 0.021). A group of Egyptian researchers showed a relationship between serum vitamin D deficiency and decreased Tregs levels in children with a physician-diagnosed cow’s milk allergy. In addition, in an in vitro study, they showed that after peripheral blood mononuclear cell (PBMC) stimulation with a cow’s milk allergen extract, this level increased when vitamin D was added [28].

Our study showed a significantly lower percentage of natural killer T cells in the group of children with vitamin D deficiency compared to the patients with normal levels of vitamin D. This dependence may be due to the fact that vitamin D is necessary for the proper development of NKT cells. As demonstrated by Cantorna et al. [26], mice with a blocked expression of VDR had a less mature state and lacked the full effector functions of iNKT cells. Similarly, Yu et al. [29] showed that vitamin D deficiency in mice fetuses resulted in NKT cell deficiency, and this deficiency could not be compensated by vitamin D supplementation later on [29]. Animal models have shown that NKT cells protect against the development of autoimmunity, and a small number of these cells in the blood is associated with a greater risk of developing autoimmune diseases [30]. In the allergy, the role of NKT cells is to maintain the Th1/Th2 balance by involving endo- and exogenous ligands for toll-like receptor 4 (TLR4) in iNKT cells [31]. There are reports showing the increased concentration of iNKT cells in bronchoalveolar lavage as well as in the peripheral blood in people with the exacerbation of bronchial asthma [32,33]. Similar observations were made in the case of atopic dermatitis. Swiss researchers showed the presence of NKT cells in biopsies collected from atopic skin lesions and places on the skin subjected to an atopy patch test (APT) [34]. On the other hand, a group of Hungarian researchers showed an impaired number and function of NKT cells in the blood of patients with atopic dermatitis compared to the control group [35]. These discrepancies may be due to the fact that the number of NKT lymphocytes may depend on the phase of the disease. These cells produce cytokines characteristic of the Th1-dependent response (IL-2, IFN-γ) and Th2 cells (IL-4, IL-5, IL-10) so they can play both protective and pathogenetic roles in allergic diseases, depending on the disease phase. It is certain, however, that their impaired number or function may disturb immunological processes, hence it is important to maintain adequate concentrations of vitamin D, which seems to be necessary for the proper functioning of these cells.

The results of our study also showed a higher concentration of IL-22 in the group of patients with vitamin D deficiency compared to patients with the optimal concentration of vitamin D in blood. IL-22 belongs to a group of cytokines produced by Th17 lymphocytes, but its source may also be Th22 lymphocytes, monocytes, macrophages and NKT cells. The essential function of this cytokine is the induction of innate immune defense in the peripheral tissue, particularly in the skin and the mucosa. IL-22 also interacts with proinflammatory cytokines. For example, it enhances the TNF-α-induced secretion of proinflammatory chemokines and molecules of the innate immune defense in keratinocytes. Therefore, the role of IL-22 in allergic diseases is not clearly established. It seems that T lymphocytes producing IL-22 are involved in the inflammatory infiltration of skin lesions in AD as well as in lung tissue in patients with asthma [36]. Similar results to our study were obtained by Kanda and colleagues, who showed that 1.25(OH)2D3 inhibits the secretion of IL-22 in patients with atopic dermatitis [37]. Another mechanism of the inflammatory process was analyzed by Anderson and colleagues. In an in vitro study, they assessed the effect of vitamin D on peripheral blood mononuclear cells (PBMCs) when stimulated with a pneumococcal whole-cell antigen (WCA). As a result, they found that VitD3 significantly reduced the Th17 cell expression of IL-17A and IL-22 [38]. An analysis of inflammation in rheumatoid arthritis also showed the beneficial effect of vitamin D in reducing the concentration of IL-22 produced by activated memory T cells [39]. Therefore, it can be concluded that vitamin D deficiency adversely affects the development of an inflammatory infiltrate dependent on the production of IL-22.

## 5. Conclusions

In summary, the results of our study confirm a positive relationship between the optimal vitamin D content in the human body and a milder course of allergic disease in children. Moreover, we showed the correlation between vitamin D deficiency and the percentage of certain lymphocytes and cytokines involved in the pathogenesis of allergic diseases.

## Figures and Tables

**Figure 1 nutrients-13-00177-f001:**
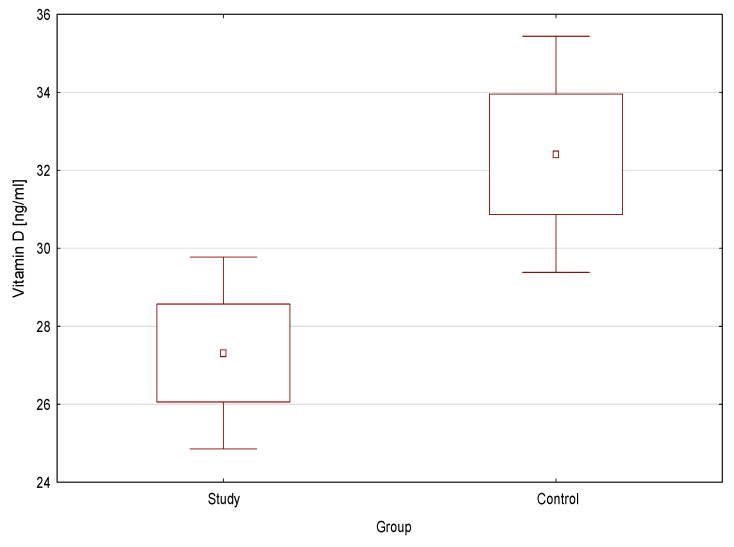
Vitamin D concentration in the study group and the control.

**Figure 2 nutrients-13-00177-f002:**
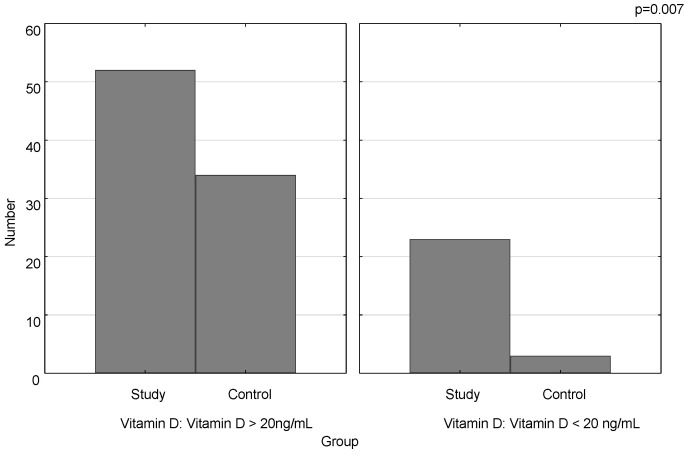
Number of vitamin D deficiency cases in the study group and the control.

**Figure 3 nutrients-13-00177-f003:**
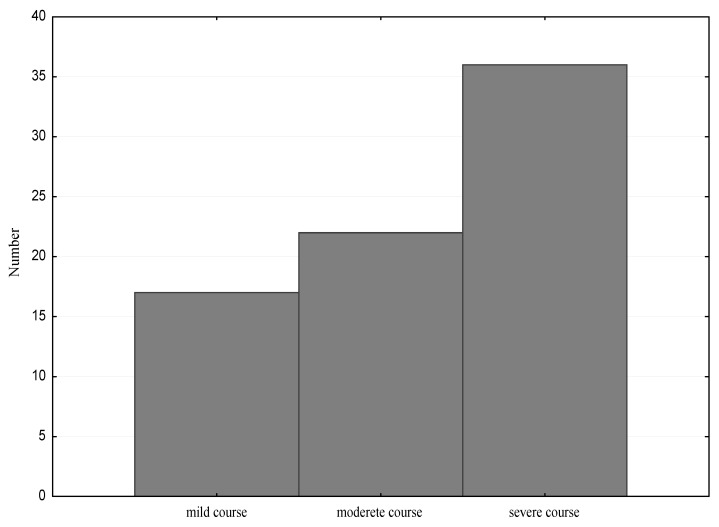
The clinical course of the allergic disease in the study group.

**Figure 4 nutrients-13-00177-f004:**
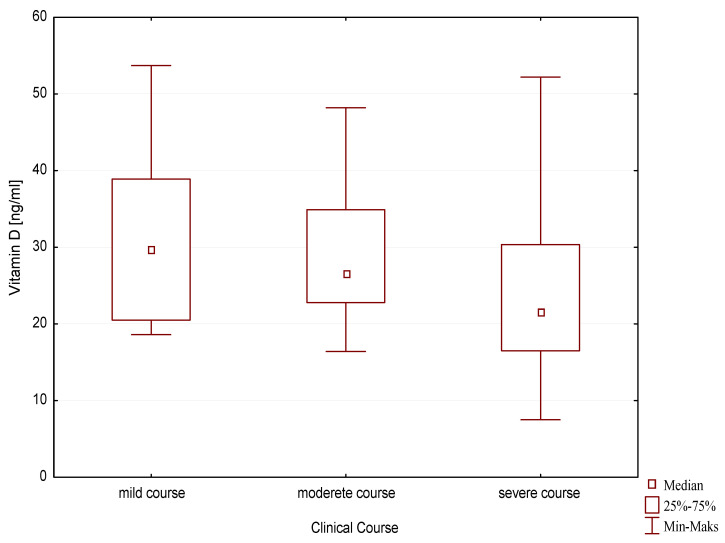
Vitamin D concentration depending on the clinical course of allergic disease.

**Table 1 nutrients-13-00177-t001:** Terminology of vitamin D concentration ranges.

Vitamin D Concentration Range	Serum 25 (OH) D Concentration	
	nmol/L	ng/mL
Severe deficiency	<25	<10
Deficiency	25–50	10–20
Insufficiency (suboptimal level)	>50–70	>20–30
Optimal level	>75–200	>30–80
Toxic level	>250	>100

**Table 2 nutrients-13-00177-t002:** Effect of vitamin D on the concentration of IgE total and eosinophils.

	Vitamin D > 20 ng/mL [Me, q_25_–q_75_]	Vitamin D < 20 ng/mL [Me, q_25_–q_75_]	*p*
IgE total [IU/mL]	329.50 (193.0–636.50)	277.00 (151.00–812.00)	ns.
Eosinophil [×10^3^/uL]	0.55 (0.29–1.04)	0.45 (0.26–0.86)	ns.
Eosinophil [%]	7.57 (4.00–9.60)	5.50 (3.00–8.90)	ns.

**Table 3 nutrients-13-00177-t003:** Effect of vitamin D on the phenotype of peripheral blood lymphocytes.

	Vitamin D > 20 ng/mL [Me, q_25_–q_75_]	Vitamin D < 20 ng/mL [Me, q_25_–q_75_]	*p*
	**Study Group**		
CD3 [%]	65.32 (63.33–69.20)	64.09 (60.29–70.10)	ns.
CD 19 [%]	15.88 (12.98–17.87)	18.13 (14.64–23.16)	ns.
CD4 [%]	35.88 (34.28–38.94)	33.85 (30.61–39.46)	ns.
CD8 [%]	28.30 (26.78–31.84)	26.31 (22.29–31.58)	ns.
CD4/CD8	1.28 (1.06–1.45)	1.33 (0.97–1.63)	ns.
CD16/56 [%]	12.61 (9.64–17.37)	10.83 (7.30–14.36)	ns.
NKT [%]	3.16 (1.45–5.23)	1.62 (0.79–3.14)	0.02
CD3 anti-HLADR [%]	5.23 (3.03–6.81)	5.38 (3.83–8.19)	ns.
T-regulatory [%]	0.76 (0.43–1.07)	0.55 (0.32–0.83)	0.05
	**Control Group**		
CD3 [%]	64.77 (60.05–68.03)	63.81 (23.34–65.17)	ns.
CD 19 [%]	15.64 (12.65–18.82)	11.63 (5.90–18.84)	ns.
CD4 [%]	34.24 (30.01–37.90)	28.59 (26.69–39.05)	ns.
CD8 [%]	26.13 (23.29–30.87)	32.54 (22.16–36.28)	ns.
CD4/CD8	1.32 (1.05–1.47)	0.82 (0.79–1.76)	ns.
CD16/56 [%]	11.61 (8.25–15.11)	19.00 (14.41–27.30)	ns.
NKT [%]	2.25 (1.27–3.27)	2.32 (2.15–4.59)	ns.
CD3 anti-HLADR [%]	4.87 (2.58–8.61)	7.53 (4.86–8.09)	ns.
T-regulatory [%]	0.87 (0.76–1.29)	0.97 (0.65–1.16)	ns.

**Table 4 nutrients-13-00177-t004:** Effect of vitamin D on blood interleukins.

	Vitamin D > 20 ng/mL [Me, q_25_–q_75_]	Vitamin D < 20 ng/mL [Me, q_25_–q_75_]	*p*
	**Study Group**		
IL-2 [pg/mL]	2.05 (0.00–3.19)	2.25 (1.09–3.13)	ns.
IL-4 [pg/mL]	1.19 (0.00–1.38)	1.42 (0.00–1.69)	ns.
IL-6 [pg/mL]	2.34 (1.81–3.23)	2.50 (1.95–3.36)	ns.
TNF-α [pg/mL]	1.53 (1.35–1.88)	1.54 (1.26–1.79)	ns.
IL-10 [pg/mL]	1.92 (1.66–2.52)	2.25 (1.83–2.93)	ns.
IFN-γ [pg/mL]	0.00 (0.00–1.19)	0.00 (0.00–1.35)	ns.
IL-17A [pg/mL]	6.07 (2.02–10.33)	5.997 (2.55–9.53)	ns.
IL-1 [pg/mL]	0.00 (0.00–0.90)	0.00 (0.00–0.96)	ns.
IL-9 [pg/mL]	2.87 (1.22–3.91)	3.01 (1.28–5.49)	ns.
IL-22 [pg/mL]	3.25 (0.00–11.47)	10.35 (3.88–34.75)	0.01
TGF-β [pg/mL]	400.53 (346.27–546.50)	419.25 (342.88–508.37)	ns.
	**Control Group**		
IL-2 [pg/mL]	1.29 (0.00–1.91)	2.25 (0.00–2.53)	ns.
IL-4 [pg/mL]	1.41 (1.14–1.69)	1.28 (0.00–1.51)	ns.
IL-6 [pg/mL]	2.18 (1.63–3.26)	4.23 (1.63–6.53)	ns.
TNF-α [pg/mL]	1.59 (1.26–2.33)	1.90 (0.00–2.37)	ns.
IL-10 [pg/mL]	1.94 (1.78–2.3)	1.52 (1.35–1.94)	ns.
IFN-γ [pg/mL]	1.30 (0.00–1.58)	0.00 (0.00–1.56)	ns.
IL-17A [pg/mL]	5.38 (2.37–7.47)	0.00 (0.00–4.91)	ns.
IL-1 [pg/mL]	0.00 (0.00–0.13)	2.01 (0.00–2.38)	ns.
IL-9 [pg/mL]	1.15 (0.01–4.80)	3.23 (0.00–4.03)	ns.
IL-22 [pg/mL]	5.39 (0.33–20.58)	0.88 (0.84–6.09)	ns.
TGF-β [pg/mL]	452.14 (358.81–535.93)	397.66 (397.04–450.20)	ns.

## Data Availability

I have read and accept Data Availability Statement.

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
