# Peer review of "Vitamin D and Immunological Patterns of Allergic Diseases in Children"

_nutrients, 2021, doi:10.3390/nu13010177_

Round 1

Reviewer 1 Report

The authors present their research work, titled: Vitamin D and immunological patterns of allergic diseases in children.

The manuscript requires extensive editing of the English language and style.

Specific comments:

Lines 48-51 and 53-55: I think some references are needed in this paragraph to support its content.

Lines 66-69:This sentence doesn't have a clear meaning and the incidence of AD and asthma does not originate from a homogeneous study population and consequently can not support the main idea.

Lines 74-76: In my view, certain editorial comments need to be addressed here.

Lines 82-83: The authors could use GINA guidelines in a more recent version, i.e 2015, the year of the begging of your study.

Lines 83-84: Atopic background is not defined only by these parameters. The authors need to differentiate clinical allergy from the atopic background and/or atopy history and document it with their study data.

Lines 86-87: According to the text, the authors refer that the patients' age ranges from6-10 and controls' age ranges from 6-11. What exactly do authors mean when they mention that patients>16 years? Besides, please clarify what and when written consent you are asking from parents. 

Line 89: Generally the authors need to define the term severity in their work. Especially in different parents of the discussion, the term 'severe allergic disease' is mentioned. How is this defined?

Lines 90-99: The authors need to analyze the definition of asthma, mild/moderate..etc, and the different steps of treatment as well. In the same line, authors need to define the initials SCORAD.

 Line 196: Could the authors support this notion with references?

Lines 200-1: The knowledge that 'children with allergies had lower average vitamin D levels compared to the group of healthy children' is well documented so far and in my view, no novelty is added here.

 Lines 202-3: The lower exposure to solar radiation is just an assumption the authors make to justify the low levels of Vit D. 

Lines 211-13: As mentioned in the methods, there is a lack of a coherent definition of the term. Here, to which exactly disease do the authors refer?

Reviewer 2 Report

The findings are of interest. Data collections, appropriately approved protocols and interpretation of the data are acceptable.

My main critique relates to background information and references. The latter are random with citation of low impact factor and not representative papers. Also some factual corrections are required.

Table listing vitamin D deficiency should be corrected as follows: severe deficiency <12 ng/ml, deficiency <20 ng/ml, insufficiency 20-30 ng/ml.

Examples of poor citations  and some inaccuracies are in the introduction specifically first three paragraphs. Citation #2 is inappropriate and not representative. Please cite paper by recognized experts in the field including reviews by Holick, Bikle and others. Paragraphs 1-3 require at least few proper citations.

Paragraph 2, from where the vitamin D1 is coming, please remove it. Vitamin D2 is formed after UVB exposure of ergosterol produced by mushrooms and plankton  (generalization for plants is incorrect). D3 is not endogenously produced it is a product of photochemical transformation of 7-dehydrocholesterol. Please correct this information and provide citations by Holick and Bikle.

Lines 45 and 46 are not fully correct since lumisterol, a photoproduct of excessive exposure of pre-D3 can be activated by hydroxylatuions with products being biologically active (Sci Rep 2017; 7:11434; Cell Biochem Biophys 78(2):165-180, 2020; Int. J. Mol. Sci. 2020, 21, 9374; doi:10.3390/ijms212), please correct this.

For activation of vitamin D3  provide  representative and proper citations.

Mention non-canonical pathways of vitamin D activation by CYP11A1 ((FASEB J 26, 3901–3915, 2012; Mol Cellular Endocrinol 383, 181-192, 2014; J Steroid Biochem Mol Biol 151,25-37, 2015; Sci. Rep. 5, 14875; doi: 10.1038/srep14875 (2015)

Mention that there are alternative to VDR nuclear receptors for active forms of D3  ((J Steroid Biochem Mol Biol. 173, 42-56., 2017; FASEB J 28:2775-2789, 2014; In J Mol Sci 2018, 19 (10), 3072)

English requires corrections by aa native English speaker.

Round 2

Reviewer 2 Report

The authors adequately replied for the most of the critique.

Some minor corrections remain in the section (lines 73-84)

CYP11A1 is also expressed in many tissues including immune cells (Genes Immun 21(3):150-168, 2020. doi: 10.1038/s41435-020-0096-6.), which should of interest to the readers.

For lines 77-80 add additional citation (Cell Biochem Biophys 78(2):165-180, 2020,  DOI: 10.1007/s12013-020-00913-6.)

Add information that active forms of vitamin D3 can also act on the arylhydrocarbone receptor (AhR)(Int J Mol Sci 2018, 19 (10), 3072; https://doi.org/10.3390/ijms19103072
